# Privy by the Bay: Emerging hotspot analysis of 311 reports of human/animal waste near San Francisco Pit Stop locations, 2009–2022

Cari A. Bogulski[1]*, William P. Watson[2], Sean G. Young[3]

1 Department of Biomedical Informatics, College of Medicine, University of Arkansas for Medical Sciences, Little Rock, Arkansas, United States of America, 2 Department of Health Policy and Management, Fay W. Boozman College of Public Health, University of Arkansas for Medical Sciences, Little Rock, Arkansas, United States of America, 3 Department of Health Data Science and Biostatistics, O'Donnell School of Public Health, University of Texas Southwestern Medical Center, Dallas, Texas, United States of America

* cbogulski@uams.edu

## Abstract

Access to basic sanitation facilities is a significant challenge for homeless populations, and evidence on the effectiveness of interventions to address this issue are still limited. San Francisco, California, has a large population of people experiencing homelessness and limited public restrooms. To increase access to public restrooms among this population in the city, the Pit Stop Program launched in 2014, introducing clean and safe public restrooms to high-need areas. This study built upon prior Pit Stop Program evaluations by conducting an Emerging Hot Spot Analysis (EHSA) using ArcGIS Pro v3.1.1 of human/animal waste reports across San Francisco to the city's 311 system using spatial and temporal characteristics, and examined the presence of these hotspots among San Francisco neighborhoods. We examined 5,940,667 reports to 311 in conjunction with Pit Stop and public restroom locations. Waste reports in San Francisco showed an upward trend from 2009 to 2022, reaching an all-time high in 2022. Pit Stop Program sites were generally concentrated in areas with the most waste reports, particularly the Tenderloin and Mission neighborhoods. Spatiotemporal hot spots were identified throughout the city. Locations containing Pit Stops were more likely to be classified as diminishing hot spots. There is evidence of improvement in the Tenderloin neighborhood, which has seen the longest and most concentrated Pit Stop intervention, however the effect appears to be small. The findings suggest the need for additional research and continued efforts to address sanitation issues for people experiencing homelessness.

## 1. Introduction

Among a host of other challenges, populations across the United States (US) [1]—including in rural areas such as Central Appalachian Kentucky [2] as well as cities

**Data availability statement:** All data are publicly available from the DataSF Open Data Portal at https://data.sfgov.org/City-Infrastructure/311-Cases/vw6y-z8j6.

**Funding:** The author(s) received no specific funding for this work.

**Competing interests:** The authors have declared that no competing interests exist.

such as New York [3], San Diego [4,5], and San Francisco [6]—continue to experience homelessness and often lack access to basic sanitation facilities, which may lead to the improper disposal of human waste. In urban areas with large populations facing homelessness, human feces can be found on sidewalks and other public spaces, creating both a nuisance and a public health concern. Numerous diseases are spread through improper sanitation, and in particular contamination with fecal matter, including cholera, hepatitis, UTIs, salmonella poisoning, and many others [7–12]. Globally, there are many barriers to relieving waste in a public setting under sanitary conditions, including an overall lack of public restrooms, poor cleanliness of public facilities, and stigma associated with using public toilets [13–17]. Within the US, the state of California continues to report some of the highest levels of homelessness in the country [18], and San Francisco reports some of the highest rates of vulnerable populations experiencing homelessness (e.g., unaccompanied youth and veterans) in the nation [19]. A 2019 report from the Pacific Institute found that San Francisco County had the highest percentage of housing without toilets of 58 counties in California [20]. Another 2019 study of public toilet availability in parks and recreational spaces in several major cities in the U.S. found San Francisco had among the fewest toilets per 100,000 residents of any of the cities studied [21]. In addition, San Francisco has a consistently large number of people experiencing homelessness. In the US Department of Housing and Urban Development's 2022 Point-in-Time (PIT) Count of people experiencing homelessness in San Francisco, 4,397 individuals were counted as unsheltered on the night of February 23, 2022, with estimates of as many as 20,000 individuals in San Francisco experiencing homelessness over the course of a year [22]. With such a large population of individuals experiencing homelessness, a lack of public restrooms creates issues for this population, who may struggle to find safe and sanitary locations to relieve waste.

To address the problem of access to safe and hygienic rest areas, San Francisco Public Works launched the Pit Stop Program in 2014, an intervention that created and maintained safe public toilets for people without consistent access to sanitation facilities [23–25]. However, evaluating the effectiveness and long-term sustainment of the Pit Stop Program is challenging, given the city-wide scope of the problem and the difficulty in measuring incidence of public waste. One data source that can be used in the evaluation of programs like the Pit Stop Program comes from 311: a reporting tool implemented in many cities including San Francisco that are designed for residents to request or access information, file complaints, submit questions, and report non-emergency issues in need of city attention over the telephone or through a smartphone app [26]. Some cities make the information reported to this system publicly available using data standards established by Open311 "to provide more actionable information for those who need it most and…[encourage] the public to be engaged with civic issues" [27]. These efforts facilitate research investigating spatio-temporal issues affecting cities, as reports are logged with timestamps and many are logged with spatial information such as address, latitude, and longitude.

Previous studies have used 311 data to forecast demand for services and investigate spatiotemporal patterns of other public health issues in areas such as New

York City, Boston, Chicago, Miami-Dade County, Columbus, and San Francisco [28–32]. Amato et al. (2022) utilized San Francisco's Open311 data to conduct an evaluation of the Pit Stop Program and several of its initiatives, such as site expansion, staffing sites, and service hour expansion using spatial analytical methods [32]. Specifically, they conducted buffer analyses surrounding Pit Stop locations using a 500-meter walking distance from a site and a six-month window prior to and following each intervention. Amato et al. (2022) found evidence for some reductions in reports of human waste for new sites and expanded service hours, largely driven by reductions in the Tenderloin, Mission, and South of Market neighborhoods.

Despite widespread recognition that public restroom access is essential to urban health and dignity, there is a striking lack of spatially grounded, empirical research evaluating the real-world impact of municipal sanitation programs in high-income cities. Most existing studies focus on access disparities or user experience, leaving a critical gap in our understanding of how interventions like San Francisco's Pit Stop program affect public hygiene outcomes over time and space. The current study aimed to build on the results of Amato et al. (2022) by conducting an Emerging Hot Spot Analysis (EHSA) of reports of human/animal waste to the city's 311 system, prior to and following the introduction of new Pit Stop Program sites to more fully characterize changes in 311 human/animal waste reports relative to Pit Stop Program expansion. We aim to: 1) geospatially characterize 311 human/animal waste reports and Pit Stop locations in San Francisco over time and 2) examine whether a reduction in 311 human/animal waste reports is observed in San Francisco neighborhoods when Pit Stop locations are introduced.

## 2. Methods

### 2.1. Data sources and processing

All data sources and relevant variables used in processing and analysis can be found in Table 1.

Data from 311 reports for San Francisco are publicly available at DataSF, and were downloaded on February 6, 2023 for the period of July 1, 2008 through February 6, 2023 [33]. The number of records downloaded, as well as the number of records included and excluded during data processing can be found in Table 2. Given the nature of 311 reports, response bias in this data set is likely (see section 4.1. Limitations).

Data were processed and analyzed using R version 4.2.1 and RStudio version 2022.2.3.492 [34,35]. After cleaning and filtering the data as described in Table 2, we analyzed spatial and temporal trends in waste reports throughout the study area, as well as partitioned by season and neighborhood (see S1 Fig for a map of San Francisco neighborhoods). To examine possible effects of seasonality, the data were additionally categorized into seasons using months: March through May were coded as Spring, June through August were coded as Summer, September through November were coded as Fall, and December through February were coded as Winter. However, no effects of seasonality were observed in the EHSA analyses, and are therefore not reported. GIS data on all 117 San Francisco neighborhoods were downloaded from DataSF (with boundaries defined by the Mayor's Office of Neighborhood Services in 2006), and total waste reports and waste report density (in thousands of reports per square kilometer) were calculated for each neighborhood.

Table 1. List of data sources and variables utilized in data processing and analysis.

| Data Set | Data Source | Variables Included for Processing | Variables Included in Analysis |
|---|---|---|---|
| San Francisco 311 Reports | DataSF | Category; Request Type; Responsible Agency; Status Notes | Latitude; Longitude; Opened (Date) |
| Pit Stop Locations | San Francisco Public Works' Pit Stop Program website | – | Address (geocoded to latitude/longitude); Year Established |
| Public Restroom Locations | OSM | tag (toilets) | Latitude; Longitude |

**Table 2. Inclusion and exclusion criteria for San Francisco 311 report data.**

| | Number of Reports | |
|---|---|---|
| **Inclusion/Exclusion Criteria** | **Included** | **Excluded** |
| 311 report data downloaded July 1, 2008 – February 6, 2023 | 5,940,667 | – |
| Included only those reports where all of the following criteria were met:<br>1) Category was "Street and Sidewalk Cleaning" or "311 External Request";<br>2) Request Type was "Human or Animal Waste" or "Human/Animal Waste" or contained "trash_porta_potties_waste_management";<br>3) and Responsible Agency did not contain "animal care," | 253,583 | 5,687,084 |
| Excluded duplicate reports using Amato et al.'s (2022) methodology by excluding any record containing the text string "dup" (short for "duplication" and related words to remove duplicated reports as identified by the Responsible Agency) in Status Notes or Responsible Agency [32] | 237,384 | 16,199 |
| Excluded reports identified as transferred to other agencies by excluding any report with the string "transfer" in Status Notes | 234,684 | 2,700 |
| Data restricted to complete calendar years (January 1, 2009 – December 31, 2022) | 230,270 | 4,414 |
| Excluded reports with any of the following:<br>1) Missing latitude or longitude<br>2) Latitude or longitude was recorded as 0<br>3) Latitude/longitude fell outside San Francisco County | **227,304** | 2,966 |

Pit Stop locations were identified from the San Francisco Public Works' Pit Stop Program website [23] and converted to a point layer based on geocoding of the intersection listed for each location. Additionally, locations with public restrooms (e.g., public libraries, bus stations, police stations, etc.) within the study area were identified and downloaded from Open-StreetMap (OSM) [36] using the "toilet" tag—indicating a toilet accessible to members of the public per OSM guidelines—and converted to a point layer. Toilets available only to customers (toilets:access = customers) were not included.

## 2.2. Data analysis

We conducted emerging hotspot analysis (EHSA) using ArcGIS Pro v3.1.1 (Esri, Redlands, CA, USA) to examine spatiotemporal trends in reports of human or animal waste near Pit Stop Program locations in San Francisco County. EHSA permits the identification of changes in trends over time, with areas categorized into one of 17 possible categories: eight "cold spot" categories, eight "hot spot" categories, and "no pattern detected" [37]. Definitions of these categories provided in the ArcGIS Pro documentation can be found in Table 3. A single point layer representing public restrooms (including Pit Stops) was created for subsequent analyses.

Geocoded waste reports were spatially and temporally aggregated into five space-time network Common Data Form (netCDF) cube data structures, which allow for the examination of human/animal waste reports at specific locations over time using spatial autocorrelation (see Sen et al., 2015, for a detailed description of the creation of and analysis using space-time netCDF cubes [38]). One space-time cube was created for all reports, and another was created for each of the four seasons. The EHSA analysis requires a pre-defined spatial unit of analysis across the study area, which was set at 1.8 hectare hexagons to represent the approximate comfortable walking distance to a restroom in time of need. No universal standard for a comfortable or appropriate walking distance to restrooms exists, although multiple recommendations have been suggested by different organizational bodies around the world for different situations and stakeholders, including 60 meters or ~0.9 hectares [39], 100 meters or ~2.6 hectares [40], and 200 feet or ~0.9 hectares [41]. Given the lack of consensus, we selected an intermediate distance of 1.8 hectares. Space-time cubes were created in ArcGIS Pro and consisted of uniform hexagonal spatial units of 1.8 hectares each and single calendar year temporal bins. EHSA uses the Getis-Ord Gi* spatial cluster detection statistic with False Discovery Rate (FDR) correction for each individual year of data, then applies the Mann-Kendall trend test to identify temporal patterns and trends across years [42–46]. Within

**Table 3. Definitions of the eight cold spot and hot spot categories and the "no pattern detected" category used in the EHSA analysis.**

| Trend Categories | Definition |
|---|---|
| New | A statistically significant cold/hot spot for the final time step and has never been a statistically significant cold/hot spot before. |
| Consecutive | Single uninterrupted run of at least two statistically significant cold/hot spot bins in the final time-step intervals, but less than 90 percent of all bins are statistically significant cold/hot spots. |
| Intensifying | A statistically significant cold/hot spot for 90 percent of the time-step intervals, including the final time step. In addition, the intensity of clustering of high counts in each time step is increasing overall and that increase is statistically significant. |
| Persistent | A statistically significant cold/hot spot for 90 percent of the time-step intervals with no discernible trend in the intensity of clustering over time. |
| Diminishing | A statistically significant cold/hot spot for 90 percent of the time-step intervals, including the final time step. In addition, the intensity of clustering in each time step is decreasing overall and that decrease is statistically significant. |
| Sporadic | A statistically significant cold/hot spot for the final time-step interval with a history of also being an on-again and off-again cold/hot spot. Less than 90 percent of the time-step intervals have been statistically significant cold/hot spots and none of the time-step intervals have been statistically significant hot/cold spots. |
| Oscillating | A statistically significant cold/hot spot for the final time-step interval that has a history of also being a statistically significant hot/cold spot during a prior time step. Less than 90 percent of the time-step intervals have been statistically significant cold/hot spots. |
| Historical | The most recent time period is not cold/hot, but at least 90 percent of the time-step intervals have been statistically significant cold/hot spots. |
| No Pattern | No statistically significant trend detected. |

hexagons containing Pit Stop locations, waste report trends and emerging hot spot categories were identified and compared against hexagons without Pit Stops using frequencies and percentages.

A weighted Pit Stop Accessibility Score was calculated to evaluate if the intensity of the Pit Stop intervention, in terms of length of time and/or spatial overlap, impacted the emerging hot spot trends for waste reports. To compute the weighted Pit Stop Accessibility Score, a score of 1 was assigned to every walking distance hexagon for each Pit Stop contained within it for every year each Pit Stop was present. For every hexagon adjacent to one Pit Stop hexagon a score of 0.5 was assigned for every year one Pit Stop was present in an adjacent hexagon. Thus, hexagons containing or adjacent to more than one Pit Stop received higher scores, and hexagons containing or adjacent to Pit Stop locations opened early in the Program would have higher scores than hexagons with more recently-established Pit Stop locations.

## 3. Results

### 3.1. Aim 1: Geospatially characterize 311 human/animal waste reports and Pit Stop locations in San Francisco over time

Human/animal waste reports in San Francisco exhibited a fairly consistent upward trend, from 5,366 reports in 2009–31,478 reports in 2022. There was a drop in reports in 2020 and 2021 (compared to 2019) during the COVID-19 pandemic, but reports nevertheless reached an all-time high in 2022. The overall upward trend, along with the drop in reports following the onset of COVID-19 in 2020 were also present in the seasonal data, with minor variations (see Fig 1).

There was also substantial spatial variation in waste reports. Overall, the four neighborhoods with the most total reports (3.4% of all neighborhoods) each had over 10,000 reports (Tenderloin, n = 39,558; South of Market, n = 39,154; Mission, n = 31,610; and Lower Nob Hill, n = 11,180), 30 additional neighborhoods (25.6% of all neighborhoods) had 1,000 or more reports (ranging from n = 1,008 in Excelsior to n = 8,536 in Civic Center), and 83 neighborhoods (70.9%) had fewer than 1,000 reports during the study time period (ranging from n = 1 in Yerba Buena Island to n = 922 in Inner Sunset). In just the most recent year of data (2022), the same top four neighborhoods each had over 3,500 reports, 42 neighborhoods (35.9%

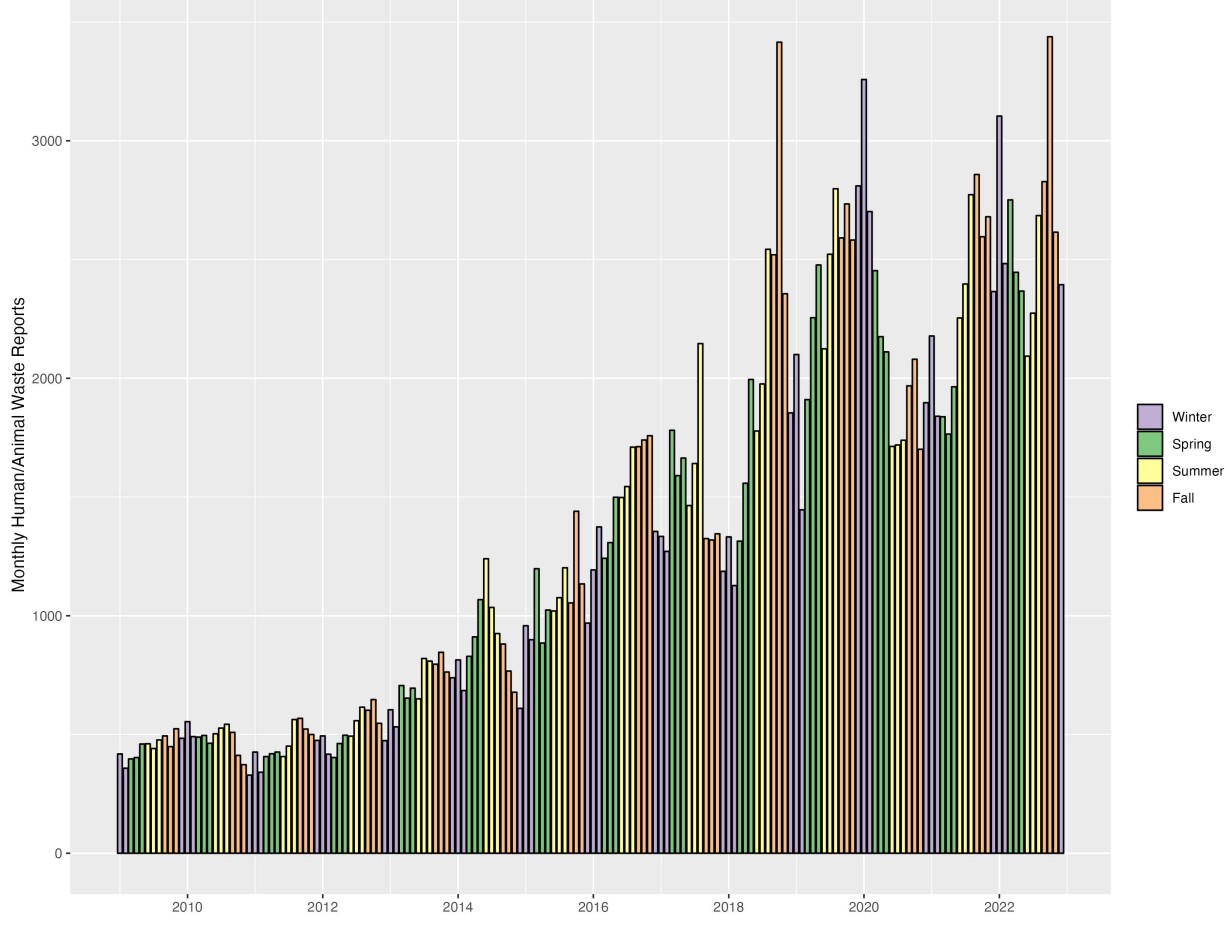

**Fig 1. Monthly pattern of human/animal waste reports in San Francisco from 2009 to 2022, colored by season.**

of all neighborhoods) had more than 100 reports, and the remaining 71 neighborhoods (60.7%) had fewer than 100 reports. Waste report density was also highest in the relatively small (a little more than half a square kilometer) Tenderloin neighborhood (74 thousand reports/sqkm), which was almost three times the density of the next highest neighborhood, Lower Nob Hill (26 thousand reports/sqkm), and nearly forty times higher than the citywide average (1.8 thousand reports/sqkm).

A total of 33 Pit Stop locations were identified, spread across 13 of the 117 total San Francisco neighborhoods. However, nearly half of all identified Pit Stop locations were concentrated in 2 neighborhoods: the Tenderloin (n = 10) and Mission (n = 6) neighborhoods. Overlaying walking-distance hexagons resulted in a total of 5,702 hexagons; of those, 32 hexagons contained Pit Stops, and an additional 134 hexagons contained other public restrooms identified from OSM. Together, less than 3% of the city's hexagons contained a public restroom, and only 15.5% of the city's hexagons either contained or were adjacent to another hexagon that contained a public restroom. Pit Stop locations were opened in different neighborhoods during different years, starting in the Tenderloin neighborhood in 2014. Fig 2 shows the trend in waste reports per year by neighborhood for the 13 neighborhoods containing Pit Stops as well as the 11 neighborhoods without Pit Stops with the highest overall counts of 311 human/animal waste reports. Neighborhoods containing Pit Stops do not exhibit a consistent trend or change in trend following Pit Stop introductions, with some examples of decreasing reports

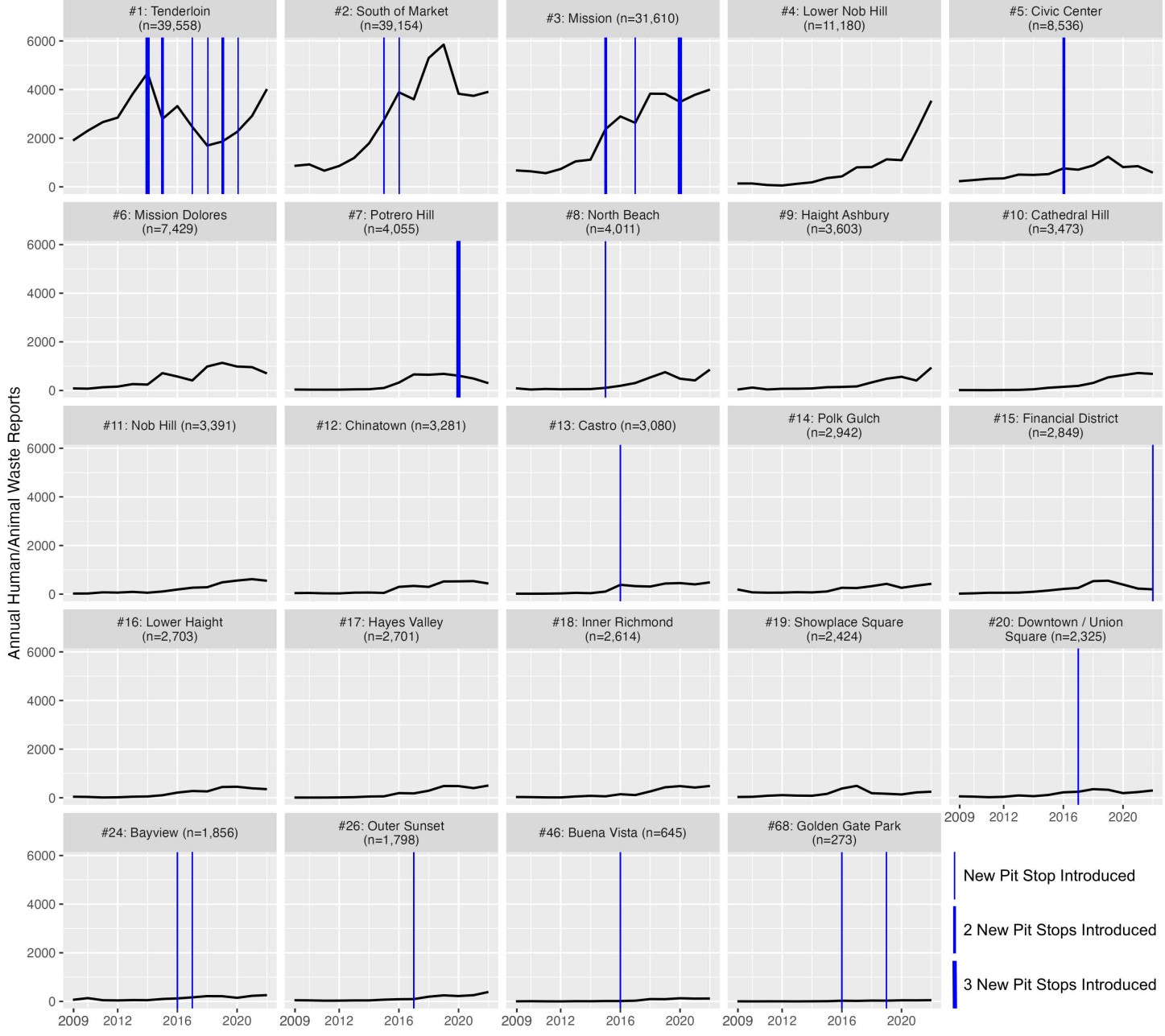

**Fig 2. Annual trends of human/animal waste reports between 2009 and 2022 for 24 selected neighborhoods, including the 13 neighborhoods containing Pit Stop locations, ranked by the total number of reports, with blue vertical lines indicating the years in which one or more Pit Stops were introduced.**

following Pit Stop introduction, and some examples of increasing reports, even within the same neighborhood in different years.

To more specifically examine and visualize the volume of 311 human/animal waste reports relative to Pit Stop location installations in time, for each Pit Stop location we calculated the total number of human/animal waste reports made

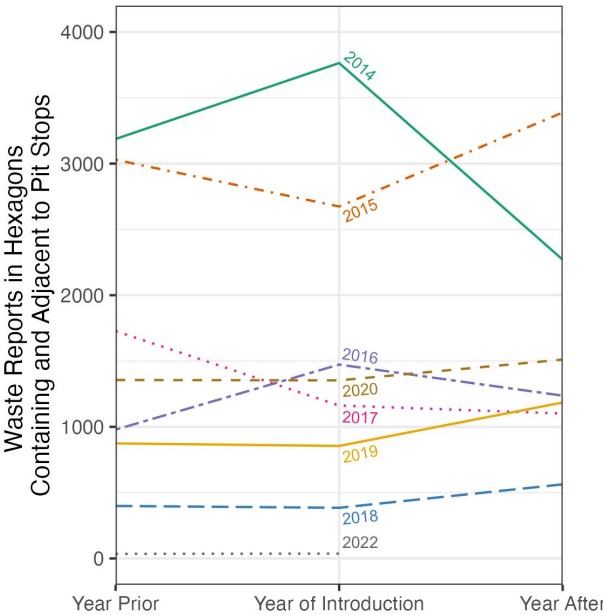

**Fig 3. 311 human/animal waste reports made in walking-distance hexagons containing and adjacent to Pit Stop locations in the years prior to, during, and after installation of a Pit Stop.**

in each location's walking-distance hexagon and its surrounding hexagons for the year prior to the location installation, the year of the location installation, and the year after the location installation (see Fig 3). As can be seen in Fig 3, only for Pit Stops installed in 2014 and 2016 was there a downward trend in reporting volume following the year of Pit Stop installation.

### 3.2. Aim 2: Examine whether a reduction in 311 human/animal waste reports is observed in San Francisco neighborhoods when Pit Stop locations are introduced

Human/animal waste reports were aggregated to the walking distance hexagons for further analysis. Fig 4A shows the total number of reports per hexagon, with Pit Stop and other public restroom locations and neighborhood boundaries over-layed for reference. Pit Stops tend to be concentrated in areas with the highest count of waste reports, most notably in the Tenderloin neighborhood. Fig 4B shows the results of the Emerging Hot Spot Analysis (EHSA) for the entire study period, with Pit Stop and other public restroom locations and city neighborhood boundaries overlayed for reference.

Most of the city (93.8% of all walking-distance hexagons) was classified as "no pattern detected" meaning either the spatial clustering tests or the temporal trend tests failed to achieve statistical significance for that location, suggesting no change in human/animal waste reporting for these areas over time. There were no cold spots (statistically-significant clusters of low report density) detected in any of the EHSA analyses. Hot spots were detected primarily in the northeastern part of the city, with one large and irregular cluster of multiple hot spot types, and nine smaller and more compact clusters. Overall, 6.2% of the city's walking distance hexagons were identified as hot spots of some type, including sporadic (1%), consecutive (3.5%), and intensifying (0.8%) hot spots, with some more limited cases of new (0.2%) and persistent (0.2%) hot spots (see Table 4). Thus, of the 352 hotspots identified, 76.1% (n = 268) were not diminishing or sporadic, meaning they were sustaining or worsening over time. There was one cluster of diminishing hot spots (n = 26, 0.5% of all walking distance hexagons and 7.4% of all hot spots) located primarily within the Tenderloin neighborhood.

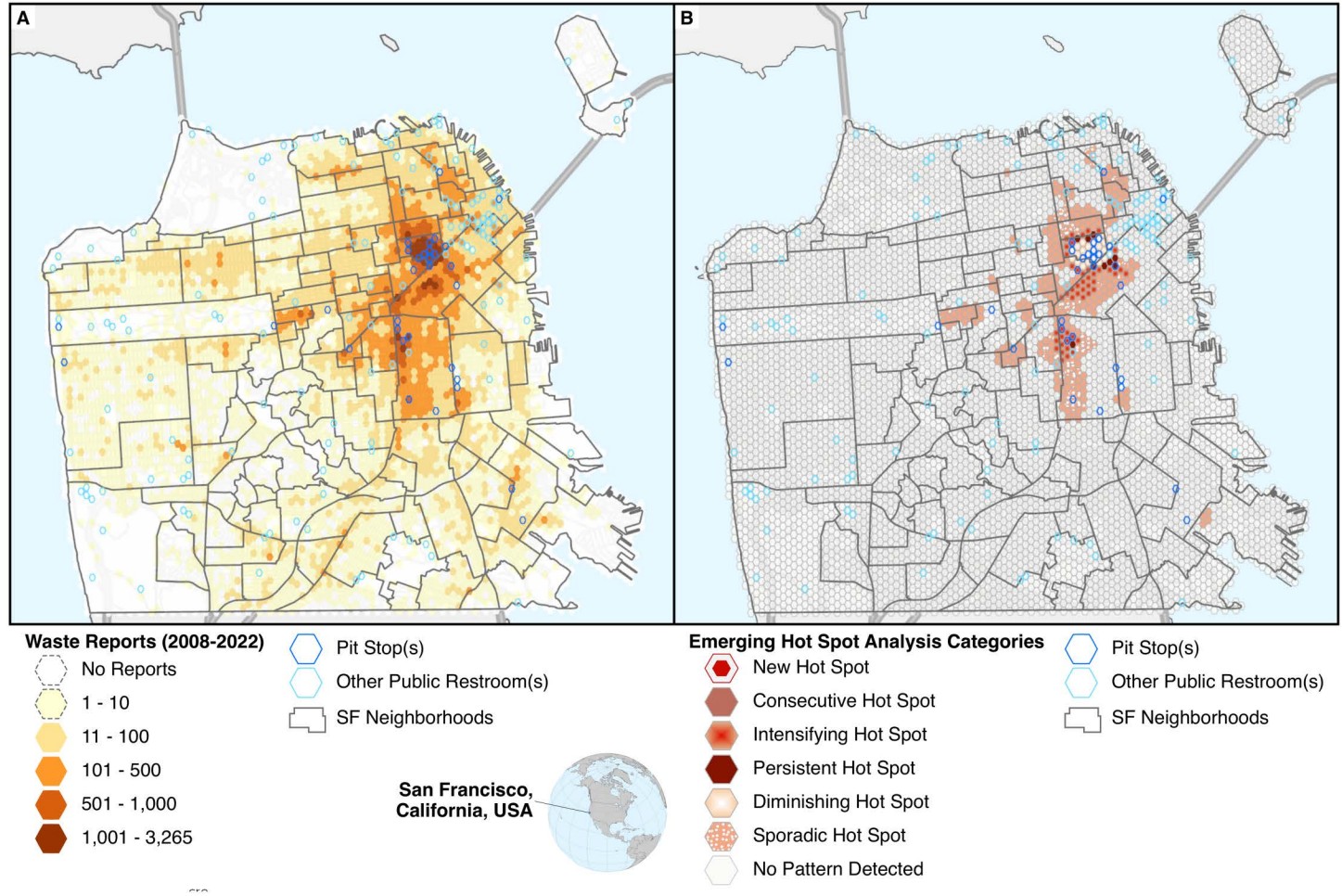

**Fig 4. Maps of San Francisco, with walking distance hexagons, Pit Stop locations, other public restroom locations, and neighborhood boundaries shown.** Map A shows the total number of human/animal waste reports from 2009 to 2022 in each hexagon. Map B shows the Emerging Hot Spot Analysis (EHSA) results for the entire study period (2009-2022).

Of the 166 walking distance hexagons containing a Pit Stop or other public restroom, 133 (80.1%) were identified as "no pattern detected." Within areas with a significant hot spot trend identified (352 of 5,702 hexagons, ~6% of the total), hexagons containing a Pit Stop (n = 21) were more likely to be classified as a diminishing hot spot (~43%) compared to hexagons without Pit Stops or other public restrooms, where only ~5% were diminishing and ~60% were classified as consecutive hot spots. Hexagons containing or adjacent to a Pit Stop accounted for 25 of 26 diminishing hot spots in the entire study area.

The Pit Stop Accessibility Score ranged from 0 for hexagons that did not contain and were not adjacent to any Pit Stop locations at any time, to 23.5 for a hexagon in the Tenderloin neighborhood containing two Pit Stops and surrounded by four others over a period of many years. The average Pit Stop Accessibility Score was 4.57 for hexagons containing or adjacent to a Pit Stop, and 0.14 for all hexagons throughout the study area. Pit Stop accessibility also varied by hot spot category, with an average score of 0 for new, 0.47 for consecutive, 0.59 for sporadic, 2.64 for intensifying, 5.06 for persistent, and 9.65 for diminishing hot spots, compared to an average score of 0.04 for no pattern hexagons. As a result, all new hot spots as defined by the EHSA did not contain or have any adjacency to an area with a Pit Stop (i.e., an average

**Table 4. Count (with column percentages) of walking-distance hexagons categorized as each defined EHSA trend, as well as average Pit Stop Accessibility Score for each EHSA trend category.**

| Trend Categories | Count (Column Percentage) of Walking-Distance Hexagons | | | | | | Average Pit Stop Accessibility Score |
|---|---|---|---|---|---|---|---|
| | Containing Pit Stop | Containing or Adjacent to Pit Stop | Containing Any Public Restroom | Containing or Adjacent to Any Public Restroom | Containing or Adjacent to Public Restrooms Excluding Pit Stops | Entire Study Area | |
| Consecutive | 4 (12.5%) | 26 (15.4%) | 11 (6.6%) | 55 (6.2%) | 30 (4.1%) | 201 (3.5%) | 0.47 |
| Persistent | 2 (6.3%) | 8 (4.7%) | 2 (1.2%) | 8 (0.9%) | 3 (0.4%) | 9 (0.2%) | 5.06 |
| Sporadic | 1 (3.1%) | 10 (5.9%) | 3 (1.8%) | 19 (2.2%) | 11 (1.5%) | 58 (1%) | 0.59 |
| Intensifying | 5 (15.6%) | 25 (14.8%) | 6 (3.6%) | 30 (3.4%) | 7 (0.9%) | 48 (0.8%) | 2.64 |
| Diminishing | 9 (28.1%) | 25 (14.8%) | 10 (6%) | 25 (2.8%) | 5 (0.7%) | 26 (0.5%) | 9.65 |
| New | 0 (0%) | 0 (0%) | 1 (0.6%) | 4 (0.5%) | 4 (0.5%) | 10 (0.2%) | 0 |
| No Pattern | 11 (34.4%) | 75 (44.4%) | 133 (80.1%) | 740 (84%) | 679 (91.9%) | 5,350 (93.8%) | 0.04 |
| TOTAL | 32 | 169 | 166 | 881 | 739 | 5,702 | |

Pit Stop Accessibility Score of 0), and that both persistent and diminishing hot spots had high average Pit Stop Accessibility Scores relative to all other hot spot categories identified.

With regard to our study aims, we observed the following:

Aim 1: Geospatially characterize 311 human/animal waste reports and Pit Stop locations in San Francisco over time

- Overall, the volume of 311 human/animal waste reports in San Francisco increased over time, without systematic influence of seasons.

- Pit Stop locations were first introduced in 2014 in the Tenderloin neighborhood, which remains the neighborhood with the largest number of Pit Stop locations anywhere in the city. The years 2015, 2016, and 2017 were the most active years for Pit Stop location installation during our study period of 2009–2022.

Aim 2: Examine whether a reduction in 311 human/animal waste reports is observed in San Francisco neighborhoods when Pit Stop locations are introduced

- 28.1% of walking-distance hexagons (n = 9) containing a Pit Stop location and 14.8% (n = 25) of hexagons containing or adjacent to a Pit Stop location had diminishing human/animal waste report hot spots.

- However, 37.5% (n = 12) of all hexagons containing Pit Stop locations were identified as consecutive, persistent, sporadic, or itensifying. Additionally, 40.8% (n = 69) of all hexagons containing or adjacent to a Pit Stop location were identified as consecutive, persistent, sporadic, or itensifying.

- No new hot spots were identified over the study period between 2009 and 2022 for hexagons containing Pit Stop Locations or for hexagons containing or adjacent to Pit Stop locations.

## 4. Discussion

The purpose of this study was to geospatially characterize 311 human/animal waste reports and Pit Stop locations in San Francisco over time and to examine whether a reduction in 311 human/animal waste reports is observed in San

Francisco neighborhoods when Pit Stop locations are introduced. We descriptively analyzed 311 human/animal waste reports and Pit Stop location installations, and conducted a city-wide EHSA to investigate whether trends in human/animal waste reporting changed over time. The most notable result of the emerging hot spot analysis (EHSA) was the cluster of diminishing hot spots centered on the Tenderloin neighborhood. The Tenderloin neighborhood had the longest and most concentrated Pit Stop intervention, containing nearly a third of the Pit Stop locations, which may explain why this area (primarily the Tenderloin neighborhood with some spillover into Civic Center, Downtown/Union Square, and South of Market) was the only part of the city with an EHSA result of diminishing hot spots. Indeed, the Hot Spot Accessibility Scores would seem to support the connection between intensity of Pit Stop intervention and improving waste report trends, with the average Accessibility Score in diminishing hot spots 9.65 compared to 0–5.06 for other hot spots trends and 0.04 for no pattern hexagons. This suggests some local impact of the Pit Stop Program on the immediately-surrounding area of a new site (see also Fig 3).

While this result seems promising, the bulk of the city failed to provide similarly optimistic results. The majority of waste reports (53.5%) were reported from four neighborhoods (3.4% of all neighborhoods), all with over 10,000 reports. Although the Tenderloin neighborhood demonstrated a large area of diminishing hotspots, the other three neighborhoods with over 10,000 reports (South of Market, Mission, and Lower Nob Hill) showed few or no areas of diminishing hotspots. Additionally, surrounding the group of diminishing hot spot hexagons concentrated in the Tenderloin neighborhood are consecutive and intensifying hot spots. South of Market, for example, which is an immediate neighbor of the Tenderloin and had the second highest total incidence of human/animal waste reports, was mostly covered with persistent or intensifying hot spots, or no pattern, as well as some consecutive hot spots, some sporadic hot spots, and a few diminishing hot spots only at the border with Tenderloin. Both the South of Market and Mission neighborhoods are much larger than Tenderloin, but had far fewer Pit Stops and large areas of consecutive, persistent, sporadic, or intensifying hot spots, which again suggests Tenderloin managed to pass some minimum effective threshold density of Pit Stops, which other neighborhoods did not exceed. Even in the Tenderloin itself, reports are trending back up in recent years (see Fig 2). Overall, despite the apparent improvement in the Tenderloin neighborhood prior to the pandemic, the spatiotemporal distribution of human/animal waste reports fails to make a compelling case for the long-term effectiveness of the Pit Stop Program. It is notable, however, that only one of the Tenderloin's 10 Pit Stop locations has been installed since 2020, with half of its Pit Stops installed between 2014–2015. A recent initiative to address this issue involves the city partnering with JCDecaux for a 20-year agreement to install and manage 25 public restrooms—including some existing Pit Stop locations—in exchange for 114 sidewalk advertising kiosks, at no cost to the city [47]. Given these recent efforts to upgrade and maintain earlier Pit Stop locations [47], it is possible that Program effectiveness has diminished as Pit Stop sites have aged without requisite maintenance. Additional work is needed to evaluate the effectiveness of these efforts to improve existing Pit Stop sites on reports of human/animal waste.

One possible explanation for the level of impact of the Pit Stop Program observed on the volume of human/animal waste reports is a possible disproportionate presence of animal waste driving these reports. One recent study observed that only 20% of fecal samples taken from San Francisco sidewalks of possible human origin (assessed visually) were, in fact, from humans [6,48]. The remaining 80% of samples were found to be of canine origin. Evidence suggests that canine waste is a larger health hazard in coastal California than human waste [49]. The data analyzed in the present study do not distinguish between human and animal waste, and thus we are unable to examine which of the reports in the 311 data were, in fact, canine rather than human in origin. Although Pit Stops include dog waste stations, it would be unsurprising to observe that an intervention designed to increase access to public toilets for humans would have a large impact on the presence of canine waste. Evidence suggests that community awareness programs such as the *There is No Poop Fairy* campaign in the Albuquerque, New Mexico [50], and other pet waste reduction campaigns informed by the Health Belief Model [51] may be successful in encouraging owners to properly dispose of canine waste. A similar program designed to reduce spread of pathogens through canine waste may also be needed as a complement to the Pit Stop Program and other similar efforts to reduce both human and animal waste on streets and sidewalks.

Many populations benefit from the expansion of access to public restrooms, including tourists, delivery drivers, and those with health conditions requiring more frequent trips to the restroom, as well as all members of a community who may be at risk for the spread of diseases when human waste is not properly disposed of [5,13,21,52]. Although a lack of public restrooms is not exclusively a problem for people experiencing homelessness, this population is especially vulnerable, given a lack of other options available to them. Additionally, although some toilets may technically be public, in practice, whether individuals experiencing homelessness feel welcome and comfortable visiting and using the toilets at these locations is a different issue [5]. Moreover, individuals experiencing homelessness disproportionately experience physical limitations that may require specific restroom accommodations that may not be available at every public toilet or Pit Stop location [53]. Although the restrooms offered by the Pit Stop Program were designed with this population in mind, the availability of other public restrooms as identified in OSM may not truly be available to all individuals experiencing homelessness.

Many Pit Stop locations align with the largest concentrations of people experiencing homelessness according to the most recent PIT counts [22]; however, there are still many areas with large concentrations of people experiencing homelessness and reports of human/animal waste. Additionally, many researchers have expressed concerns about the consistency of PIT count data, as well as the bias of undercounting less visible types of homelessness, such as temporarily staying with others or living in a vehicle [54,55]. There may also be influences of local, regional, state-wide, and federal policies—including anti-camping ordinances, anti-sleeping ordinances, and ordinances prohibiting sitting or lying down in public places—enacted or repealed during the study period, as many such policies have affected people in San Francisco experiencing homelessness [56]. Thus, better estimates of both incidents of improperly disposed human waste and census of people experiencing homelessness are needed to better understand the impact of interventions such as the Pit Stop Program.

## 4.1. Limitations

Several limitations of this study warrant discussion and future research. First, as described above, evidence suggests that canine waste may be a larger public health problem in San Francisco than human waste [6,48], but 311 reports of waste cannot distinguish between human and animal samples. Different data collection methods are needed to isolate the incidence and prevalence of human waste on streets and sidewalks, which would facilitate a more precise analysis of the impact of such (human) public restroom interventions. Second, finding high quality data on public restrooms within the city is a problem recognized by many of the citizens living there [57]. Also, as stated above, identification of a public restroom in OSM does not necessarily mean that it is truly available and accessible to people experiencing homelessness, in the face of "sanitation injustice" [5] and disproportionately higher rates of physical disabilities [53] that may require accommodations not available at all public restrooms. Additionally, OSM is a collection of publicly-sourced data, and may not be comprehensive or always accurate in identifying restrooms available to the public. Although OSM guidelines encourages tagging restrooms in the database only if they are accessible to the public, we were unable to verify the accuracy of particular restrooms identified in the OSM point layer we accessed. In addition, for the purposes of our analysis, the public restrooms identified through OSM lacked temporal information, limiting us to summary observations under the possibly flawed assumption that the identified restrooms were present and accessible throughout the study period. Data reflecting a more comprehensive public restroom census, including dates of operation, would be necessary to examine the temporal influence of other public restrooms.

We also lacked data on Pit Stop utilization and did not differentiate Pit Stops by new restroom installations compared to the new provision of attendants at existing public restrooms. There are also challenges with the human/animal waste reports from 311, including the aggregation or indistinguishability of human and animal waste, as well as a lack of information on the number of users generating reports. It is possible, for example, that a small number of individuals living in specific parts of the city may make disproportionate use of the 311 system to report human or animal waste. Such massively

unequal reporting behaviors could heavily skew the waste report density in certain areas of the city, which may bias our findings. Some research has been conducted on waste samples taken directly from the sidewalks of San Francisco based on reports made in the 311 system [6,48], avoiding reporting bias. However, these studies have necessarily included much smaller samples over shorter periods of time, but nevertheless could perhaps inform models of human/animal waste data mining.

Finally, we anticipated an impact of the COVID-19 pandemic on our results; however, other than an overall decrease in the number of human/animal waste reports in 2020 and the first few months of 2021, our general pattern of results remained the same including or excluding data collected during the pandemic. People without homes in San Francisco experienced higher rates of mortality during the first year following the declaration of the COVID-19 public health emergency, which could potentially explain this decrease [58]. Additionally, as homeless shelters in San Francisco reported COVID-19 outbreaks [59,60], the city implemented an Alternative Shelter Program, moving ~450 people from shelters to hotel rooms and providing other facilities such as trailers and Safe Sleep tent sites to prevent the spread of the virus [61,62]. Several Pit Stop locations were added during this time, but were later removed. These additional Pit Stop locations may have pushed Accessibility Pit Stop score past a minimum threshold, thereby contributing to a reduction in waste incidents. As an alternative explanation, to combat the spread of COVID-19, San Francisco and later the state of California issued several executive orders addressing allowable activities beginning in March 2020, including business restrictions and other pandemic-related policies and behaviors, which may have reduced the number of 311 reports and biased the data toward undercounting incidents of human/animal waste [63]. The observed drop in waste reports in 2020 would support either scenario (see Fig 1). To combat these possible data discrepancies, our EHSA utilized annual data to de-emphasize some of this short-term variability; however, data quality issues remain and our results must thus be interpreted with caution.

## 5. Conclusion

Overall, reports of waste to San Francisco 311 increased from 2009 to 2022, and 5.7% (n = 326) of the walking-distance hexagons across 37 of the city's 117 neighborhoods (32%) exhibited new, consecutive, persistent, sporadic, or intensifying hotspot trends. In the Tenderloin area specifically, which has been a focus of the Pit Stop Program since its inception in 2014, there is some limited evidence that Pit Stops may help reduce human/animal waste reports in their immediate vicinity, but the overall effect appears to be small, and may require a minimum density of Pit Stop locations in an area to be effective. Additionally, recent evidence suggests that much of the waste on the sidewalks of San Francisco are canine in origin. To address this problem, additional interventions, such as pet waste reduction campaigns, may be necessary to more comprehensively reduce the improper disposal of human/animal waste. Given the high cost of the Pit Stop Program and the limited evidence of positive impact on reducing waste reports, our results can be used to inform where future Pit Stop (or similar program) locations are most needed to improve service efficiency, and at what density given areas of high volumes of human/animal waste reports. Additionally, alternative programs and supplemental interventions should be explored.

## Supporting information

**S1 Fig. Neighborhoods of San Francisco, California, United States.**
(PNG)

## Acknowledgments

We are grateful to Heather K. Amato, Douglas Martin, Christopher M. Hoover, Jay P. Graham, and Avery Richards for sharing their data cleaning scripts and overall process with our team.

## Author contributions

**Conceptualization:** Sean G. Young.

**Data curation:** Cari A. Bogulski.

**Formal analysis:** Cari A. Bogulski, William P. Watson, Sean G. Young.

**Investigation:** Cari A. Bogulski, William P. Watson, Sean G. Young.

**Methodology:** Cari A. Bogulski, William P. Watson, Sean G. Young.

**Project administration:** Cari A. Bogulski, Sean G. Young.

**Supervision:** Sean G. Young.

**Validation:** William P. Watson, Sean G. Young.

**Visualization:** Sean G. Young.

**Writing – original draft:** Cari A. Bogulski, William P. Watson, Sean G. Young.

**Writing – review & editing:** Cari A. Bogulski, William P. Watson, Sean G. Young.

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
