## [Decision Letter · Decision Letter 0]

PONE-D-25-03015Privy by the Bay: Emerging hotspot analysis of 311 reports of human/animal waste near San Francisco Pit Stop locations, 2009-2022PLOS ONE

Dear Dr. Bogulski,

Thank you for submitting your manuscript to PLOS ONE. After careful consideration, we feel that it has merit but does not fully meet PLOS ONE’s publication criteria as it currently stands. Therefore, we invite you to submit a revised version of the manuscript that addresses the points raised during the review process.

We look forward to receiving your revised manuscript.

Kind regards,

Clement Ameh Yaro, Ph.D

Academic Editor

PLOS ONE

Journal Requirements:

2. We note that Figure 3 in your submission contain [map/satellite] images which may be copyrighted. All PLOS content is published under the Creative Commons Attribution License (CC BY 4.0), which means that the manuscript, images, and Supporting Information files will be freely available online, and any third party is permitted to access, download, copy, distribute, and use these materials in any way, even commercially, with proper attribution. For these reasons, we cannot publish previously copyrighted maps or satellite images created using proprietary data, such as Google software (Google Maps, Street View, and Earth). For more information, see our copyright guidelines: http://journals.plos.org/plosone/s/licenses-and-copyright.

a. You may seek permission from the original copyright holder of Figure 3 to publish the content specifically under the CC BY 4.0 license.  

Reviewers' comments:

Reviewer's Responses to Questions

**Comments to the Author**

1. Is the manuscript technically sound, and do the data support the conclusions?

Reviewer #1: Partly

Reviewer #2: Partly

Reviewer #3: Partly

Reviewer #4: Yes

Reviewer #5: Yes

Reviewer #6: Yes

2. Has the statistical analysis been performed appropriately and rigorously? 

Reviewer #1: Yes

Reviewer #2: N/A

Reviewer #3: Yes

Reviewer #4: Yes

Reviewer #5: Yes

Reviewer #6: Yes

3. Have the authors made all data underlying the findings in their manuscript fully available?

Reviewer #1: Yes

Reviewer #2: Yes

Reviewer #3: Yes

Reviewer #4: Yes

Reviewer #5: Yes

Reviewer #6: No

4. Is the manuscript presented in an intelligible fashion and written in standard English?

Reviewer #1: Yes

Reviewer #2: Yes

Reviewer #3: Yes

Reviewer #4: Yes

Reviewer #5: Yes

Reviewer #6: Yes

5. Review Comments to the Author

Reviewer #1: Please see attached comments.

Overall

I think this is a well-conducted study. The main concern with the article is that there’s now convincing evidence that most of the fecal matter on SF sidewalks may not be from humans, but rather from canines. This needs to be more fully addressed to better identify the best policies to reduce fecal matter on sidewalks.

Abstract

Consider adding public toilet or public sanitation to key words

Introduction

I think it’s important to highlight that much of the fecal waste is potentially from dogs. News sources suggest high levels of dog ownership among the unhoused population: article. More importantly, a Science of the Total Environment study looked at human markers in fecal samples collected from sidewalks in SF and found that most of the fecal waste on SF sidewalks is likely to be from canines: https://doi.org/10.1016/j.scitotenv.2024.170139. In that study, the authors set out to collect human feces only by selecting larger fecal samples (i.e. likely to be human based on size), but only a small fraction of the samples were identified as human.

Methods

Can you describe more about how public restrooms, other than Pit Stops, were identified and labeled? Fire departments vs libraries, etc.

Results

Line 134. Can you add the range of reports for the 34 neighborhoods with over 1,000 reports?

Line 131-135. There’s mention of 117 neighborhoods, but the categories discussed are 4 neighborhoods with lots of report; 34 neighborhoods with over 1,000 reports and 7 with less than 1 report per year. What about the other 72 neighborhoods?

Lines 131-138. Can you also add the annual reports by neighborhood categories in the most recent year?

Discussion

Can you discuss how certain public restrooms may affect individuals’ willingness to use them? For example, in fire departments or police departments that have public toilets, I doubt people know about them or feel comfortable using them.

Line 205. For this sentence (“While this result seems promising, the bulk of the city failed to provide similarly optimistic results.”), I think there may need to be additional information added here. It seems important to mention that most of the reports occur in 4 neighborhoods (percent?).

Additionally, I think it’s important to recognize that interventions for dog feces management may require novel approaches, and public toilets won’t likely solve that problem. Please add literature on what interventions might work if this is mostly canine feces.

Lines 281-282. I think “much of the city” is a mischaracterization; I’d suggest that you give some estimate of the neighborhoods (e.g., one-third of the neighborhoods had…) in each of the categories: persistent, sporadic and intensifying.

There are a lot of benefits of public toilets (e.g., emergency situations for Uber and Amazon drivers), which are missing in the discussion. See: “Legislative influence on the decline of public toilet provision in the UK: a case study of Leeds city” DOI: 10.2166/washdev.2024.380

Reviewer #2: Thank your for your excellent work. The manuscript has some potential to create a new database and identify the major spatial areas of interventions for waste management but needs streamlining. This manuscript has used an emerging tool which pave the way of application of such tool in varied way that is appreciable. I want to see a well-structured and coherent manuscript with fundamental chronology such as Introduction, Literature review/conceptual framework, Methods and Tools (Study area, tools, data, data processing, analysis process), Results, Discussion, Conclusion. With the revision, I believe, the authors can contribute more to the research community. Detail comments are attached.

Reviewer #3: The two most significant issues with this paper are identified by the authors in the limitation’s sections of their paper. However, I feel that they are underestimating the impact of these two factors which are that waste data in the 311 reports does not differentiate between human or animal waste and that restroom locations from Open Street Map have no temporal data. I feel that to be suitable for publication this paper needs to more conclusively demonstrate that animal waste is not a statistically significant portion of the 311 reports or demonstrate that variable does not affect the results. Not knowing the dates at which different rest room facilities became available is less of an issue but should also be addressed. Ideally the authors would demonstrate that the spatial patterning and availability of restrooms facilities has not changed in significant ways over the time period that this study examines, or changes would not affect the results. Two lesser issues are that some statements lack clear sourcing, and that data filtering and cleaning procedures are not explained, perhaps this latter point could be addressed through supplementary material. Otherwise, the analysis is sound, relevant, and of interest.

Other minor comments are as follows:

1) The 311 system should be explained in greater detail, particularly for international audiences who may not have had exposure to this system of reporting issues to civil government.

2) There are a few statements that require additional sourcing (lines 46, 47).

3) The referenced San Francisco neighborhoods should be displayed on a map as many audiences will not be familiar with this local geography.

4) Data cleaning and filtering operations should be described to allow for replicability of this analysis (line 87).

5) Sourcing should be provided to justify the use of a 1.8 hectare walking distance hexagons (Iine 107).

6) In line146 additional justification should be provided for why the specific 11 neighborhoods without Pit Stops were selected for including in the comparison in Figure 1.

7) Lines 224-233 could more directly address the implications of comparing the Pit Stop program budget to other homelessness programs.

8) I fail to see the relevance of lines 243-250, the connection of this paragraph to the rest of the paper should be elucidated or this section removed.

9) I am unfamiliar with the “Karen effect” on line 260-262 and this should explained and sourced.

10) Fig. 3 would benefit from zooming in on the Northeast portion of the city which contains the hotspots.

Reviewer #4: This is a empirically solid paper! I commend the authors for providing a clearly-documented analytical approach, one which, to the extent possible, takes into account the obvious limitations to the study. (The coincidence of the pandemic with the study period is notable, as is the difficulty of controlling for the density of people experiencing homelessness.) I would propose several revisions, which should not present substantial barriers to incorporation.

First, I note the large number of hexagonal bins containing no reports of human or animal waste. This is a likely source of bias in the study's results, and could lead to either Type I or Type II errors. (Intuitively, I would also imagine that the lack of cold spots follows from the inflated zero count.) The authors should consider reporting results of an analysis in which these observations are dropped (potentially even including cases where there was only one report over the relatively long study period, which would seem to be a strong indicator that this is a case of dog-walkers more than unhoused people).

Secondly, and relatedly, I think the analysis would be more compelling if the authors performed the analysis of only a subset of neighborhoods where there are known to be relatively large numbers of people experiencing homelessness. This would also have the effect of addressing the same zero-inflation bias as the above. With these first two suggestions, my basic point is that the study, as it stands, is a comparison of, essentially, areas where Pit Stop restrooms have been installed and all other areas in the city. The more compelling study is a comparison of areas where Pit Stop restrooms have been installed and areas experiencing at least broadly comparable levels of homelessness.

Finally, as an addition to the limitations section: in addition to disruptions brought about by the pandemic, I would think that, in many parts of the city, incidents of human waste would be substantially effected by changes in San Francisco's approach to policing homelessness, breakups of encampments, etc. (not to mention how it approaches street-cleaning). While this is hard to capture empirically, it's bears mentioning.

Reviewer #5: The article shows high quality in terms of background introduction, research purpose, data processing and analysis results. However, there is still room for improvement in some aspects.

In the description of the research background, the literature review is relatively sufficient, but more discussion can be made on the international experience of similar urban health interventions to enhance the universality of the research. The uniqueness of San Francisco in the health issues of homeless people can be further emphasized, such as the special challenges and policies in homeless management and health facility provision in San Francisco compared with other cities. In addition, the previous research results and limitations of the Pit Stop project can be briefly reviewed to more clearly show the innovation and development of this study.

The reported data may be limited by the reporting habits of residents (for example, some areas may have insufficient data due to fewer residents or low social attention). Although this is mentioned in the discussion section, it is recommended to add a description of the data limitations and possible coping strategies in the method section.

The conclusion section is basically consistent with the research findings, but the core contributions of the article should be further summarized and policy recommendations should be highlighted. For example, it can be discussed how the Pit Stop project can optimize site selection or improve service efficiency to enhance the practical value of the research.

When analyzing the results, further exploration could be conducted into why improvements were seen in some areas (such as the Tenderloin) but not in others. For example, differences between areas could be considered, such as homeless population density, program coverage, and community engagement.

Overall, the study has high publication value and is recommended for publication after revision.

Reviewer #6: Review summary

The manuscript involves an Emerging Hot Spot Analysis of 311 reports for human/animal waste in San Francisco, between 2009 and 2022 building upon the Pit Stop Program and previous work that has been conducted concerning the analysis of these spots, both spatially and temporally. The authors state the problem successfully, presenting the situation in San Francisco compared to other areas of the USA and explain the Pit Stop Program as well how the 311 reports are collected. The research gap is not fully described and needs further explanation. The methods used in the manuscript are well described and the methods used are appropriate. There are some minor issues with the description of data as well as the clarification of categories of hot spots used in the research work. The results are presented in a sophisticated way and the subject has been approached quite well. The use of graphs and table is helpful and informative. The results present some a few minor issues. The discussion are developed in an holistic way, including points that have not been mentioned before in the previous sections, such the “medical waste”. It is important that the limitations section has been thoroughly described. The map and graphs presented are very well made. It is important the authors have added supplementary material. They have not added the full list of reports used but this is not a big issue as these data are publicly available, as it is mentioned in the manuscript. According to my opinion, the manuscript can be published after minor changes.

Major issues

In my opinion, there are no major issues concerning the manuscript.

Minor issues

1. The research gap that can be covered by the manuscript should be better developed.

2. Probably the exact number of neighborhoods can be mentioned (line 97).

3. How was the comfortable walking distance to a restroom in time calculated? If it comes from a published work then this can be mentioned.

4. It would be better if the categories of spots (lines 111, 112) are described, even with an example.

5. The description of neighborhood results (lines 131-138) is a bit vague, making it difficult to follow.

6. Even at the end of the results section, the connection between pit stop points and neighborhoods was unclear, according to my opinion.

7. Even though I find the maps used very presentable, informative and clear, I would suggest their separation in two different maps as they are not very well visible in one page.

8. Table 1 is clear and well made. However, it was unclear to me how the average pit stop accessibility score was used later in the manuscript. It seems that it is not sufficiently developed although it is a very useful information.

9. It would be probably useful for the “medical waste” section to be described in the methods section and not be presented for the first time in the discussions.

10. Concerning the limitations, they are very well developed. Nevertheless, could the authors describe any actions taken concerning these limitations? It is not clear to me but this is a section that probably not many actions can be taken anyway.

11. The full list of reports and data could be presented but as the authors mentioned these data are open and accessible so this is not a problem.

12. The supplementary graph is very clear and informative. I would recommend to be added in the main body of the manuscript but this is just a recommendation.

13. It would be really great if the readers could read a section on how this study and generally studies like this can contribute to better implementation of Pit Stop Programs.

Positive issues

• This is a research work with a detailed spatial analysis, useful for both scientific and social policy issues.

• The maps and graphs of the manuscript are created in a sophisticated way using R and GIS, adding up to the quality of the work.

• The introduction section might not lengthy but it contains everything that is needed for the reader to understand what is included in the research (the selection of San Francisco, the 311 reporting, etc.), only lacking the gap description.

• The description of the selection of reports and data is clear and well developed.

• Both the spatial and temporal analysis are very well approached, as well as neighborhood comparisons, despite some points being unclear.

• Limitations are very well described.

6. PLOS authors have the option to publish the peer review history of their article (what does this mean? ). If published, this will include your full peer review and any attached files.

**Do you want your identity to be public for this peer review?** For information about this choice, including consent withdrawal, please see our Privacy Policy .

Reviewer #1: No

Reviewer #2: **Yes: ** Tafsirul Islam

Reviewer #3: No

Reviewer #4: No

Reviewer #5: No

Reviewer #6: **Yes: ** Aikaterini Christopoulou

---

## [Author Response · Author response to Decision Letter 1]

5 Jun 2025

We have uploaded a Word document entitled "Response to Reviewers" as a separate file, which contains responses to all comments.

---

## [Decision Letter · Decision Letter 1]

Privy by the Bay: Emerging hotspot analysis of 311 reports of human/animal waste near San Francisco Pit Stop locations, 2009-2022

PONE-D-25-03015R1

Dear Dr. Bogulski,

We’re pleased to inform you that your manuscript has been judged scientifically suitable for publication and will be formally accepted for publication once it meets all outstanding technical requirements.

Kind regards,

Clement Ameh Yaro, Ph.D

Academic Editor

PLOS ONE

Additional Editor Comments (optional):

Reviewers' comments:

Reviewer's Responses to Questions

**Comments to the Author**

1. If the authors have adequately addressed your comments raised in a previous round of review and you feel that this manuscript is now acceptable for publication, you may indicate that here to bypass the “Comments to the Author” section, enter your conflict of interest statement in the “Confidential to Editor” section, and submit your "Accept" recommendation.

Reviewer #2: All comments have been addressed

Reviewer #3: All comments have been addressed

2. Is the manuscript technically sound, and do the data support the conclusions?

Reviewer #2: Yes

Reviewer #3: Yes

3. Has the statistical analysis been performed appropriately and rigorously? 

Reviewer #2: Yes

Reviewer #3: Yes

4. Have the authors made all data underlying the findings in their manuscript fully available?

Reviewer #2: Yes

Reviewer #3: Yes

5. Is the manuscript presented in an intelligible fashion and written in standard English?

Reviewer #2: Yes

Reviewer #3: Yes

6. Review Comments to the Author

Reviewer #2: Thank you for your intriguing revision. The authors have addressed all the comments point by point. The revision looks good to me. Best of luck!

Reviewer #3: Thank you for your careful and detailed consideration of comments on this paper. The added clarification around terminology, San Francisco geography, and data cleaning procedures alleviate my previously stated concerns. Particularly changs in the discussion and limitations sections address issues around the temporality of washrooms in OSM data and the ratio of human to animal waste.

7. PLOS authors have the option to publish the peer review history of their article (what does this mean? ). If published, this will include your full peer review and any attached files.

**Do you want your identity to be public for this peer review?** For information about this choice, including consent withdrawal, please see our Privacy Policy .

Reviewer #2: **Yes: ** S. M. Tafsirul Islam

Reviewer #3: No

---

## [Editor Report · Acceptance letter]

PONE-D-25-03015R1

PLOS ONE

Dear Dr. Bogulski,

I'm pleased to inform you that your manuscript has been deemed suitable for publication in PLOS ONE. Congratulations! Your manuscript is now being handed over to our production team.

Kind regards,

on behalf of

Dr. Clement Ameh Yaro

Academic Editor

PLOS ONE